# Adopting and validating a technology acceptance model-based paradigm to assess acceptance and satisfaction with electronic health information system by healthcare providers in resource-limited governmental and non-governmental hospitals

Ahmad Dweikat[1,2], Nidal Dwaikat[3]*, Ramzi Shawahna [4,5]*

1 Engineering Management Master Program, Faculty of Graduate Studies, An-Najah National University, Nablus, Palestine, 2 An-Najah National University Hospital, Nablus, Palestine, 3 Department of Industrial Engineering, Faculty of Engineering and Information Technology, An-Najah National University, Nablus, Palestine, 4 Department of Physiology, Pharmacology, and Toxicology, Faculty of Medicine and Allied Health Sciences, An-Najah National University, Nablus, Palestine, 5 Clinical Research Center, An-Najah National University Hospital, Nablus, Palestine

* nidal_n@najah.edu (ND); ramzi_shawahna@hotmail.com, ramzi.shawahna@najah.edu (RS)

## Abstract

In resource-limited healthcare settings, adoption of electronic health information systems (EHIS) depends on provider acceptance and satisfaction. This study extended the Technology Acceptance Model (TAM) with Technology-Organization-Environment (TOE) factors to capture contextual influences in Palestinian hospitals. A cross-sectional survey of 220 healthcare professionals (December 2023-June 2024) assessed eight TOE constructs, including relative advantage, system quality, compatibility, complexity, top management support, IT support/training, and competitive pressure, alongside TAM beliefs (perceived usefulness, perceived ease of use) and behavioral intention. Partial least squares structural equation modeling (PLS-SEM) confirmed reliability and validity (indicator loadings 0.76-0.96; Cronbach's $\alpha$ 0.72-0.91; AVE 0.64-0.91; SRMR 0.075-0.079). The model explained 68% of variance in perceived usefulness, 57% in ease of use, and 74% in behavioral intention. Both ease of use ($\beta = 0.43$, $p < 0.001$) and usefulness ($\beta = 0.39$, $p < 0.001$) significantly predicted intention. System quality positively influenced usefulness ($\beta = 0.30$, $p < 0.001$) and ease of use ($\beta = 0.23$, $p < 0.001$). Competitive pressure was strongly associated with both constructs (PU: $\beta = 0.23$, $p = 0.004$; PEOU: $\beta = 0.37$, $p < 0.001$). Complexity showed no significant effect. Hierarchical regression confirmed TOE-TAM constructs explained 51–76% additional variance beyond demographics. The validated TAM-TOE instrument demonstrated robust psychometric properties and strong explanatory power for EHIS acceptance in resource-constrained contexts. Key levers for implementation include enhancing system quality, leveraging competitive

**Data availability statement:** All relevant data are contained within the paper and its Supporting Information files.

**Funding:** The author(s) received no specific funding for this work.

**Competing interests:** The authors have declared that no competing interests exist.

**Abbreviations:** AVE: Average variance extracted; BI: Behavioral intention; COMP: Compatibility; CP: Competitive pressure; CPLX: Complexity; CR: Composite reliability; EHIS: Electronic health information systems; HTMT: Heterotrait-Monotrait; IRB: Institutional review board; ITS: IT support and training; PEOU: Perceived ease of use; PLS-SEM: Partial least squares structural equation modeling; PU: Perceived usefulness; RA: Relative advantage; SD: Standard deviation; SQ: System quality; SRMR: Standardized root mean square residual; STROBE: Strengthening the reporting of observational studies in epidemiology; TAM: Technology acceptance model; TMS: Top management support; TOE: Technology-organization-environment; UNRWA: United Nations Relief and Works Agency.

pressure, aligning EHIS with workflows, and reinforcing organizational support. This tool provides evidence-based guidance to accelerate EHIS uptake, optimize clinical processes, and improve patient outcomes.

## Author summary

Electronic health information systems (EHIS) are increasingly important for improving patient care, but their adoption in resource-limited hospitals remains challenging. This study explored the factors that shape healthcare providers' acceptance and satisfaction with EHIS in Palestinian governmental and non-governmental hospitals. We extended the well-known Technology Acceptance Model (TAM) by integrating organizational and environmental factors from the Technology-Organization-Environment (TOE) framework. A survey of 220 healthcare professionals was conducted, and advanced statistical modeling was used to test how system quality, compatibility with workflows, management support, training, and competitive pressure influence perceptions of usefulness, ease of use, and intention to adopt EHIS. The findings showed that ease of use and perceived usefulness were the strongest predictors of adoption. System quality and competitive pressure also played important roles, while complexity did not significantly affect perceptions of usefulness—likely due to participants' extensive experience with digital systems. Overall, the extended TAM–TOE model explained a large proportion of adoption behavior and demonstrated strong reliability. These results highlight practical strategies for policymakers and hospital leaders to accelerate EHIS uptake, optimize clinical workflows, and ultimately improve patient outcomes in resource-constrained settings.

## Introduction

Electronic Health Information Systems (EHIS) promise to revolutionize healthcare delivery by securing patient records, streamlining clinical workflows, and supporting data-driven decision-making [1,2]. EHIS comprise integrated digital platforms for patient record management, clinical decision support, laboratory and pharmacy workflows, and real-time data reporting [1–6]. In high-resource settings, these platforms have demonstrably reduced medication errors, accelerated diagnostics, and bolstered patient safety [1–7]. However, in resource-limited environments, characterized by intermittent power, unreliable network connectivity, and limited onsite information technology (IT) support, EHIS adoption rates remain low in many low- and middle-income hospital settings [3,8]. Palestine, for example, has deployed the AviCenna EHIS across governmental hospitals and the electronic registry of the United Nations Relief and Works Agency (UNRWA) at primary-care centers [9], yet clinician engagement remains limited, perpetuating paper-based workarounds and undermining system sustainability. Many users cite steep learning curves, insufficient training, and

misaligned workflows as reasons for under-utilization [3,8]. Resistance to change, coupled with concerns about increased documentation burdens, can erode clinician buy-in and stall digital transformation efforts [10,11]. These practical challenges not only slow clinical workflows but also perpetuate reliance on paper records, with the attendant risks of misplaced files, transcription errors, and disrupted care during emergencies.

Given the clear benefits and persistent obstacles, it is essential to unpack the factors associated with healthcare providers' acceptance and satisfaction with EHIS. A nuanced understanding of both individual perceptions and organizational/environmental conditions will inform targeted interventions to boost uptake and optimize system performance. To this end, we build on the Technology Acceptance Model (TAM), which foregrounds perceived usefulness (PU) and perceived ease of use (PEOU), and extend it with Technology-Organization-Environment (TOE) factors to capture contextual influences unique to low-resourced hospitals.

In this study, we focus on EHIS acceptance among professionals with clinical (physicians, nurses, and pharmacists) and non-clinical roles (laboratory and radiology technicians, and medical secretaries) working in both governmental and nongovernmental Palestinian hospitals. By examining the full spectrum of end users and integrating established theoretical frameworks with local realities, we aim to develop actionable insights that might drive safe, efficient, and sustainable EHIS adoption.

The TAM, introduced by Davis in 1989 [12], posits that two core beliefs determine an individual's intention to adopt a new technology: PU and PEOU. PU reflects the extent to which users believe a system will enhance their job performance, while PEOU captures the degree of effortlessness associated with system operation. Although TAM has been extensively applied in healthcare, to study electronic health records, clinical decision support systems, and other digital tools, it often explains only moderate proportions of variance in adoption behavior when used in isolation [13]. This suggests that individual attitudes, while important, do not fully account for the complexity of technology uptake in clinical environments.

Complementing TAM, the TOE framework broadens the analytical lens to include organizational readiness and environmental forces [14]. Technology factors encompass system quality (SQ), compatibility (COMP) with existing workflows, and perceived complexity (CPLX). Organizational factors cover top management support (TMS) and IT support and training (ITS). Environmental factors mainly involve competitive pressure (CP). By integrating TAM and TOE, researchers can capture both micro-level beliefs and macro-level contexts, yielding a more complete picture of factors associated with adoption.

Building on these foundations, our extended TAM-TOE paradigm incorporates seven additional constructs tailored to resource-constrained hospitals [15–18]: relative advantage (RA), COMP, CPLX, TMS, ITS, SQ, and CP. Each factor is linked to PU and PEOU, which in turn influence behavioral intention (BI). This integrated model offers a robust theoretical scaffold for testing how individual perceptions interact with institutional and environmental determinants to shape EHIS adoption.

Several investigations have applied extended TAM-TOE models to healthcare contexts, demonstrating that augmenting PU and PEOU with factors like management commitment, IT infrastructure, and external mandates can improve explanatory power, sometimes accounting for up to 65% of the variance in usage intentions in high-income settings [13–16,19]. However, when these hybrid frameworks are deployed in resource-constrained environments, they tend to underperform, revealing that domain-specific variables remain unmodeled. Previous TAM–TOE studies in low- and middle-income countries have often reported relatively modest explanatory power. For example, Sulaiman et al. (2023) found that the model explained 42% of variance in PU and 55% in BI [20]. Other studies in low- and middle-income countries reported suboptimal $R^2$ values [1,3–6,9,12,15–18,21–41], underscoring the need for extended frameworks that capture additional contextual predictors. These figures are notably lower than those typically observed in high-income settings, underscoring the need for extended frameworks that capture additional contextual predictors in resource-limited environments.

## Aim of this study

This study aims to develop and validate a context-sensitive, integrated TAM-TOE paradigm for assessing EHIS acceptance and satisfaction among diverse healthcare professionals in Palestinian governmental and nongovernmental hospitals. We seek to operationalize key technology factors (RA, SQ, COMP, and CPLX), organizational factors (TMS and ITS), environmental pressure (CP), and core TAM constructs (PU, PEOU, and BI), then test their structural relationships using partial least squares structural equation modeling (PLS-SEM).

## Model description and hypotheses

Drawing on an extensive literature review, we propose an integrated framework ([Fig 1]) in which seven exogenous drivers feed into PU and PEOU, which in turn influence BI. Specifically:

### Relative advantage (RA)

Defined as the extent to which EHIS improves service quality, productivity, and cost or time efficiency, RA is hypothesized to bolster PU (H1).

H1: RA positively influences PU. Healthcare providers who believe EHIS offers clear improvements in service quality and efficiency will view it as more useful.

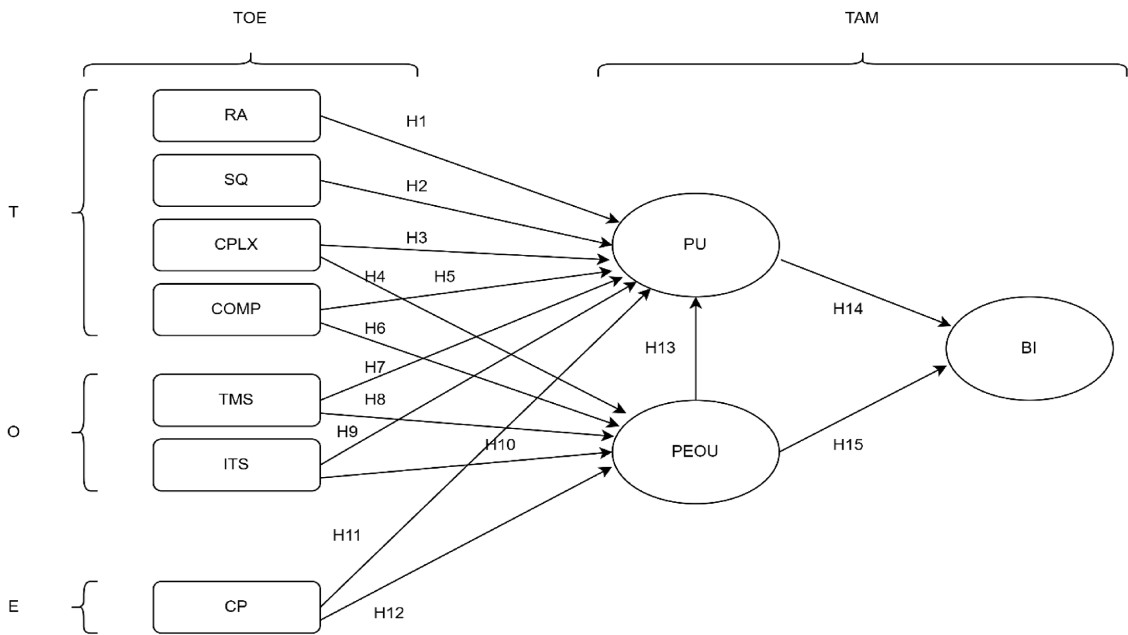

**Fig 1. Conceptual paradigm integrating the TAM and the TOE framework.** This model positions seven exogenous constructs, four technological factors (RA, SQ, COMP, and CPLX), two organizational factors (TMS and ITS), and one environmental factor (CP), as antecedents to the core TAM beliefs of PEOU and PU. Arrows H1-H12 represent hypothesized paths from each TOE driver into PU and PEOU. PEOU further enhances PU (H13), and both PEOU and PU exert direct, positive effects on BI to adopt EHIS (H14-H15). The integrated TAM-TOE paradigm thus illustrates how individual perceptions and broader contextual forces jointly shape healthcare professionals' readiness to use electronic health information systems. BI: behavioral intention, COMP: compatibility, CP: competitive pressure, CPLX: complexity, ITS: IT support and training, PEOU: perceived ease of use, PU: perceived usefulness, RA: relative advantage, SQ: system quality, TMS: top management support.

### System quality (SQ)

Encompassing system accuracy, stability, and user-friendly interface design, SQ is expected to enhance both PU (H2) and PEOU (H3).

H2: SQ positively influences PU. Systems that are accurate, stable, and user-friendly enhance users' perceptions of utility.

H3: SQ positively influences PEOU. High system reliability and interface design reduce effort required to use EHIS.

### Complexity (CPLX)

CPLX captures users' perceptions of technical difficulty and mental effort required to operate EHIS. We hypothesize a negative effect of CPLX on PU (H4).

H4: CPLX negatively influences PU. Greater perceived system CPLX undermines users' belief in its usefulness.

### Compatibility (COMP)

The degree to which EHIS aligns with existing IT infrastructure, workflows, and job requirements constitutes COMP. We posit that COMP positively influences PU (H5) and PEOU (H6).

H5: COMP positively influences PU. When EHIS is well aligned with existing workflows and IT infrastructure, healthcare providers are more likely to perceive it as valuable.

H6: COMP positively influences PEOU. A good fit with current work practices lowers the learning curve and effort.

### Top management support (TMS)

Visible commitment from leadership, including resource allocation and strategic alignment, is theorized to increase both PU (H7) and PEOU (H8).

H7: TMS positively influences PU. Visible leadership commitment and resource allocation bolster users' sense of system value.

H8: TMS positively influences PEOU. Management-led training and clear guidance reduce perceived effort.

### IT support and training (ITS)

The availability of formal training, technical assistance, and resources is hypothesized to strengthen PU (H9) and PEOU (H10).

H9: ITS positively influence PU. Adequate technical assistance and formal training enhance users' judgments of EHIS utility.

H10: ITS positively influence PEOU. Hands-on training and responsive helpdesk services minimize operational difficulties.

### Competitive pressure (CP)

Awareness of peer institutions' EHIS adoption and perceived market advantage are hypothesized to positively associated PU (H11) and PEOU (H12).

H11: CP positively influences PU. Awareness of peer institutions' EHIS adoption highlights potential performance gains.

H12: CP positively influences PEOU. External benchmarking and industry norms motivate users to learn and use EHIS effortlessly.

### Perceived ease of use (PEOU)

PEOU reflects the perceived effortlessness of EHIS operation. We hypothesize that higher PEOU will both elevate PU (H13) and directly encourage BI (H15).

H13: PEOU positively influences PU. Systems that are easier to operate are also judged to be more valuable.

H15: PEOU positively influences BI. Lower effort requirements foster higher willingness to adopt EHIS.

### Perceived usefulness (PU)

PU indicates the belief that EHIS enhances job performance. We hypothesize that greater PU will be associated with stronger BI (H14).

H14: PU positively influences BI. Greater perceived benefits are associated with stronger intentions to use EHIS in clinical practice.

### Behavioral intention (BI)

BI represents healthcare professionals' readiness to employ EHIS in their daily practice. It is modeled as the ultimate outcome of PU and PEOU in our integrated paradigm.

By empirically testing these hypotheses among clinical (physicians, nurses, and pharmacists) and non-clinical (laboratory and radiology technicians, and medical secretaries) professionals, this study will illuminate the multifaceted determinants of EHIS acceptance and satisfaction in a severely resource-constrained healthcare setting.

## Methods

### Study design and settings

The study employed a cross-sectional design in adherence to the STrengthening the Reporting of OBservational studies in Epidemiology (STROBE) statement (S1 Table). Data were collected from different governmental and nongovernmental hospitals across the West Bank of Palestine. The diverse hospital settings included in the study should have enhanced the generalizability of our findings across different types of healthcare institutions within the region.

Data were collected from both governmental and nongovernmental hospitals across the West Bank of Palestine. This diversity of hospital settings enhances the generalizability of our findings across different types of healthcare institutions in the region. Governmental hospitals in the West Bank are connected through the AviCenna Health Information Medical System, implemented by the Palestinian Ministry of Health to unify electronic records across facilities [9]. Private hospitals employ similar EHIS platforms. These contextual details underscore the complexity of EHIS adoption in Palestine: while governmental hospitals benefit from a unified platform, private and UNRWA facilities operate on parallel systems. This fragmentation highlights the importance of consensus on essential EHIS features and the need for interoperability to ensure that digital transformation efforts translate into meaningful improvements in patient care.

### Participants

Healthcare professionals were recruited from governmental and nongovernmental hospitals across the West Bank of Palestine between December 2023 to June 2024. Eligible participants included professionals with clinical (physicians, nurses, and pharmacists) and non-clinical roles (laboratory technicians, radiology technicians, and medical secretaries) who had at least six months of experience using an EHIS.

Sample size justification was based primarily on PLS-SEM–relevant criteria. The sample size needed for this study was calculated using the 10-Times Rule, that requires at least 10 times the maximum number of structural paths pointing at any one latent construct. In our model, the greatest number of incoming paths is eight (to PU), yielding a minimum sample size of:

$$n = 10 \times 8 = 80$$

Again, we cross-validated our target sample against that calculated using the inverse square root method [42]. For a medium effect size ($f^2 = 0.15$), $\alpha = 0.05$, and desired statistical power ($1 - \beta = 0.80$), the following formula was used to calculate the needed sample size:

$$n = \left(\frac{Z_{1-\frac{\alpha}{2}}}{f^2}\right)$$

This formula yielded a sample size of 171 participants.

We then used the Raosoft sample size calculator ([www.raosoft.com](www.raosoft.com)) to calculate the sample size for a finite population of approximately 1,200 healthcare staff. Assuming a 5% margin of error, a 90% confidence level, and a 50% response distribution, Raosoft yielded a required sample size of 220 participants. The Raosoft formula is based on a standard proportion estimate:

$$n = \frac{N \times Z^2 \times p(1-p)x^2}{(N-1)E^2 + Z^2 \; P(1-p)}$$

where $N$ is the population (1,200), $Z$ is the z-score for 90% confidence (1.645), $p$ is the assumed proportion (0.5), and $E$ is the margin of error (0.05).

As the Raosoft calculation yielded the larger requirement, we adopted 220 as our recruitment target. To account for potential non-response, we distributed 250 questionnaires using purposive sampling, targeting healthcare professionals with at least six months of EHIS experience.

An interviewer-administered questionnaire was used to collect data. Eligible participants were identified via each hospital's human resources department and approached in person by the main investigator, who explained the study's objectives. After obtaining written informed consent, the interviewer recorded demographic and practice-related information and guided each participant through the questionnaire items.

## Measurement scales

The study employed a structured questionnaire encompassing ten latent constructs, each measured with multiple items on a five-point Likert scale (1 = strongly disagree to 5 = strongly agree). The questionnaire was developed based on established TAM and TOE constructs, adapted to the EHIS context. Table 1 summarizes the constructs, item codes, and brief item descriptions.

The RA construct comprised four items, while SQ included five items. COMP and CPLX were each represented by four items. Organizational factors, TMS and ITS, were measured with four items apiece. CP was captured by three items from the same source. Core TAM beliefs of PU and PEOU each featured six items, and BI was assessed via four items. These items were adapted from previous studies [1, 3-6, 9, 12, 15-18,21-41].

## Translation, cultural adaptation, and pilot testing

Given the study's Palestinian context, the English-language questionnaire was translated into Arabic using a forward-backward procedure [43]. Two independent bilingual experts produced initial Arabic translations, which were then back-translated into English by a third expert. Discrepancies between the original and back-translated versions were reconciled in a consensus meeting involving all translators and the research team. The finalized Arabic questionnaire was reviewed by two senior healthcare researchers to ensure semantic equivalence and contextual sensitivity before data collection commenced.

Prior to full deployment, the instrument underwent qualitative pretesting through interviews with 15 healthcare professionals (5 physicians, 5 nurses, and 5 administrative staff) to evaluate item clarity, relevance, and cultural appropriateness. Feedback from these sessions led to the rewording of three items to improve readability and the removal of one redundant item. Subsequently, a pilot test was conducted with 27 healthcare providers across two hospitals; all scales demonstrated acceptable internal consistency (Cronbach's α ≥ 0.78) [44], confirming the questionnaire's reliability for the target population.

## Procedure

The questionnaire consisted of two parts. In the first part, demographic and practice-related data were recorded, including sex, educational qualifications, age, years of hospital experience, duration of EHIS/computer use, formal EHIS/computer

**Table 1. Summary of constructs and measurement items.**

| Construct | Code | Item |
|---|---|---|
| Relative advantage (RA) | RA1 | EHIS improves efficiency compared to paper-based processes |
| | RA2 | EHIS enhances quality of patient care |
| | RA3 | EHIS provides better access to patient information |
| Complexity (CPLX) | CPLX1 | EHIS is difficult to learn and operate |
| | CPLX2 | EHIS requires considerable effort to use |
| Compatibility (COMP) | COMP1 | EHIS fits well with existing workflows |
| | COMP2 | EHIS aligns with current IT infrastructure |
| | COMP3 | EHIS is consistent with professional practices |
| System quality (SQ) | SQ1 | EHIS is reliable and stable |
| | SQ2 | EHIS provides accurate information |
| | SQ3 | EHIS responds quickly during use |
| Top management support (TMS) | TMS1 | Hospital leadership actively supports EHIS use |
| | TMS2 | Adequate resources are allocated for EHIS |
| IT support (ITS) | ITS1 | Training is available for EHIS users |
| | ITS2 | Technical support is accessible when needed |
| Perceived ease of use (PEOU) | PEOU1 | EHIS is easy to use |
| | PEOU2 | Learning to operate EHIS is straightforward |
| Perceived usefulness (PU) | PU1 | EHIS improves job performance |
| | PU2 | EHIS enhances productivity |
| Behavioral intention (BI) | BI1 | I intend to continue using EHIS |
| | BI2 | I would recommend EHIS to colleagues |

EHIS: electronic health information system

training, and job title. In the second part, participants answered items designed to operationalize constructs from the TAM and related frameworks: RA, COMP, CPLX, TMS, ITS, SQ, PU, PEOU, BI, and CP. Specific items for each construct are detailed in S2 Table, and the full questionnaire is provided in S3 Table.

## Data analysis

**Data preprocessing and preliminary analysis.** Data management began with the organization and coding of demographic variables (e.g., age, sex, years of experience, educational background) using IBM SPSS version 21.0. In this initial phase, descriptive statistics were generated, and normality was confirmed via skewness and kurtosis tests. Specifically, skewness values within the range of -2 to +2 and kurtosis values between -7 and +7 substantiated that the data distribution was approximately symmetric with acceptable tail behavior [45]. Establishing normality at this stage provided confidence that subsequent analyses would not be unduly influenced by severe non-normal distribution.

**PLS-SEM measurement model evaluation.** For the core latent constructs, PLS-SEM was executed using SMART-PLS 4. This method was chosen due to its flexibility with non-normal data and its ability to model complex relationships among latent variables. The evaluation began with the outer model (or measurement model), where the focus was on ensuring that each observed indicator robustly reflected its underlying theoretical construct. Key evaluation criteria included: outer loadings, internal consistency reliability, convergent validity, and discriminant validity. For the outer loadings, each indicator's loading on its latent construct was examined. Loadings closer to ± 1 suggest a stronger relationship, and a conventional threshold of 0.70 was applied to ascertain indicator reliability. Indicators that did not meet this threshold were earmarked for removal or revision.

All constructs were specified as reflective, aligning with their theoretical conceptualization as latent variables manifested through observed indicators. We evaluated this reflective measurement model in SmartPLS by examining indicator loadings; assessing internal consistency reliability using Cronbach's α and composite reliability (both exceeding 0.70) [46, 47]; confirming convergent validity via average variance extracted (AVE > 0.50); and establishing discriminant validity through the Fornell-Larcker criterion [48, 49]. Additionally, discriminant validity was also assessed using the heterotrait-monotrait (HTMT) ratio of correlations, with values ≤ 0.90 indicating adequate discriminant validity. These analyses showed that indicator loadings met recommended thresholds, reliability coefficients indicated consistent scales, AVE values demonstrated that constructs captured the majority of indicator variance, and discriminant validity criteria were satisfied.

### Model refinement process

During the model evaluation phase, an iterative refinement process was undertaken to strengthen measurement reliability and validity. Indicators with low outer loadings (< 0.70), inadequate internal consistency (Cronbach's α < 0.70), or insufficient convergent validity (AVE < 0.50) were systematically removed. In the pilot study, the CPLX→PEOU path was initially specified in accordance with TAM literature. However, preliminary analyses revealed instability in this relationship and very poor performance of one PEOU indicator (PEOU4 = 0.043). To preserve the integrity of the measurement model, problematic indicators were deleted and the CPLX→PEOU path was excluded from the final specification. This refinement aligns with established recommendations that up to 20% of indicators may be removed to improve model fit and reliability [50]. These adjustments ensured that the retained constructs were psychometrically robust and that the final model provided a valid representation of the data.

### PLS-SEM structural model assessment

Once the measurement model was validated, attention shifted to the inner (structural) model, which examined the relationships among latent variables. The following steps were undertaken: path analysis and bootstrapping, predictive power, $f^2$, and multicollinearity diagnostics. For the path analysis and bootstrapping, direct, indirect, and total effects among constructs were estimated. Bootstrapping with 5,000 resamples was conducted in SmartPLS to generate robust standard errors and assess the statistical significance of the path coefficients. A 95% confidence interval was derived from the bootstrap distributions, and a fixed random-number seed was applied to ensure full reproducibility. A path coefficient of 0.1 or above, coupled with a $P$-value < 0.05, was deemed indicative of a meaningful relationship. For the predictive power, the coefficient of determination ($R^2$) for endogenous constructs was calculated. $R^2$ values exceeding 0.20 were interpreted as demonstrating acceptable predictive power for the model. The $f^2$ statistic was computed to gauge the practical impact of each exogenous variable on the dependent constructs. This metric evaluates how removal of an individual predictor affects the $R^2$, thereby confirming the substantive relevance of each hypothesized relationship. For the multicollinearity diagnostics, the variance inflation factor was used to check for collinearity among predictor variables. Variance inflation factor values < 5 provided assurance that multicollinearity was not compromising the stability and interpretation of the regression estimates.

### Overall model fit assessment

To determine how well the complete model fit the underlying data, the standardized root mean square residual (SRMR) was calculated. An SRMR value of less than 0.1 was used as the cutoff for a good fit, indicating that the discrepancies between observed and predicted covariances were acceptably small. This step was critical for confirming that the refined measurement and structural models together provided a reliable and valid representation of the data.

### Hierarchical regression analyses

To complement the PLS-SEM findings and assess incremental variance contributions, hierarchical regression analyses were performed in SPSS 21.0. Variables were entered in sequential blocks: Block 1: control variables (training, job title (role), and years of experience) and Block 2: Core predictors corresponding to latent constructs identified in the PLS-SEM model. At each step, change in $R^2$ ($\Delta R^2$) was examined to determine the additional variance explained by new associated factors. Significance of $R^2$ was tested via F-change statistics, and standardized beta coefficients were reported to compare predictor strengths. Multicollinearity was again monitored via variance inflation factor values (< 5). This hierarchical approach elucidated both the unique and incremental explanatory power of key theoretical variables beyond demographic controls.

### Ethical considerations

The study was conducted in accordance with international and local ethical principles governing scientific research, including those in the Helsinki Declaration. Ethical approval was obtained from the Institutional Review Board (IRB) of An-Najah National University (approval #: IRB Mas.Dec.2023/12). Additionally, approvals were secured from the competent authorities and management at each participating hospital. Prior to participation, all participants provided written informed consent, and stringent measures were implemented to ensure both the privacy of the participants and the confidentiality of the collected data throughout the study.

## Results

### Characteristics of the participants

Of the 250 potential participants initially invited, 220 responded and were included in the analysis, yielding a response rate of 88% (S1 Fig). Among the 220 participants, 55.0% (n = 121) were male and 45.0% (n = 99) were female, with a mean age of 32.8 years (SD = 6.2). Most participants were aged 30–39 years. Regarding educational level, the majority held an undergraduate degree (70.9%, n = 156), while 22.3% (n = 49) had postgraduate qualifications and 6.8% (n = 15) had a diploma. Of the participants, 161 (73.2%) had clinical roles and 59 (26.8%) has non-clinical roles. Participants reported an average of 8.6 years (SD = 5.0) of work experience. Furthermore, 45.9% (n = 101) had received formal training on EHIS, whereas 54.1% (n = 119) had not, and the mean duration of EHIS or computer usage was 10.4 years (SD = 6.0). The detailed demographic and practice variables of the participants are shown in Table 2.

### Measurement model evaluation

**Reliability and convergent validity.** To ensure internal consistency and indicator stability, Cronbach's α and composite reliability were evaluated using the PLS-SEM to assess the relationships between observed indicators and their latent constructs. We adopted a minimum threshold of 0.70 for both metrics, in line with established psychometric standards. This evaluation confirms that the indicators accurately represent the underlying theoretical concepts. In addition, validity was assessed through convergent validity (AVE ≥ 0.50), discriminant validity (using the Fornell-Larcker criterion, HTMT ≤ 0.90, and cross-loadings), and overall model fit (using the standardized SRMR with a cut-off of < 0.08 to indicate good fit).

Table 3 presents a consolidated overview of our measurement model's psychometric properties. This high level of reliability and validity ensures that our constructs can be interpreted with confidence, minimizing measurement error in subsequent structural analyses. All indicator loadings exceed the 0.70 benchmark (range: 0.760 to 0.960), confirming individual item reliability. Construct-level internal consistency is strong, with Cronbach's α values ranging from 0.72 to 0.91 and composite reliability from 0.84 to 0.96, both well above the recommended 0.70 threshold. Convergent validity is likewise supported, as (AVE values fall between 0.64 and 0.91, exceeding the 0.50 criterion. Together, these results verify that each construct is measured reliably and captures adequate variance from its indicators.

PLOS Digital Health

**Table 2. Demographic and practice variables of the participants.**

| Variable | n (%) or mean (SD) |
| --- | --- |
| **Sex** | |
| Female, n (%) | 99 (45.0) |
| Male, n (%) | 121 (55.0) |
| **Age (years), mean (SD)** | **32.8 (6.2)** |
| **Age (years)** | |
| 20 to 29, n (%) | 70 (31.8) |
| 30 to 39, n (%) | 116 (52.7) |
| 40 to 49, n (%) | 32 (14.5) |
| 50 to 59, n (%) | 2 (0.9) |
| **Educational level** | |
| Diploma, n (%) | 15 (6.8) |
| Undergraduate degree, n (%) | 156 (70.9) |
| Postgraduate degree, n (%) | 49 (22.3) |
| **Specialty/job title** | |
| Non-clinician, n (%) | 59 (26.8) |
| Clinician, n (%) | 161 (73.2) |
| **Years of experience, mean (SD)** | **8.6 (5.0)** |
| **Received training on the EHIS** | |
| No, n (%) | 119 (54.1) |
| Yes, n (%) | 101 (45.9) |
| **Duration of EHIS/computer usage (years), mean (SD)** | **10.4 (6.0)** |

EHIS: electronic healthcare system, SD: standard deviation

High outer loadings signified that the observed indicators were strongly correlated with their latent variables, thereby capturing a substantial proportion of the common variance. Because the outer loadings ranged from -1 to +1, items with loadings closer to the absolute value of 1 indicated a more robust relationship between the indicator and its respective construct. These robust loadings further justify retaining all indicators and underscore the measurement model's stability across diverse respondent profiles.

### Discriminant validity

Discriminant validity was first examined via the Fornell-Larcker criterion (square root of AVE on the diagonal vs. inter-construct correlations off-diagonal). As shown in Table 3, every diagonal entry exceeds its corresponding off-diagonal correlations, confirming that each construct shares more variance with its own indicators than with any other construct. This pattern demonstrates that our latent variables are empirically distinct, thereby reducing potential multicollinearity concerns in the structural model. The combined matrix is shown in Table 4.

To provide a more robust assessment, we calculated the full HTMT matrix (Table 5). HTMT values ranged from 0.1 to 0.9. All values were ≤ the conservative threshold of 0.90. Combined with Fornell–Larcker and cross-loading results, these findings confirm the empirical distinctiveness of each construct and alleviate any concerns about overlap.

### Model diagnostics

Multicollinearity among predictor variables can lead to unstable regression estimates and inflated standard errors, thereby compromising the reliability of the model's outcomes. To address this concern, the variance inflation factor was computed for

**Table 3. Item loadings, means, standard deviations, Cronbach's α, composite reliability, and average variance extracted of the initial measurement model.**

| Construct | Item | Loading | M | SD | α | CR | AVE |
|---|---|---|---|---|---|---|---|
| **RA** | RA1 | 0.86 | 3.45 | 0.82 | 0.87 | 0.91 | 0.66 |
| | RA4 | 0.86 | 3.52 | 0.79 | | | |
| | RA5 | 0.79 | 3.30 | 0.85 | | | |
| | RA6 | 0.76 | 3.28 | 0.88 | | | |
| | RA7 | 0.81 | 3.41 | 0.81 | | | |
| **SQ** | SQ1 | 0.85 | 3.67 | 0.74 | 0.72 | 0.84 | 0.64 |
| | SQ2 | 0.79 | 3.59 | 0.79 | | | |
| | SQ3 | 0.76 | 3.55 | 0.82 | | | |
| **COMP** | Comp1 | 0.82 | 3.21 | 0.91 | 0.86 | 0.91 | 0.78 |
| | Comp2 | 0.93 | 3.30 | 0.89 | | | |
| | Comp3 | 0.90 | 3.27 | 0.94 | | | |
| **CPLX** | CPLX1 | 0.95 | 2.10 | 1.02 | 0.90 | 0.92 | 0.80 |
| | CPLX2 | 0.83 | 2.33 | 0.98 | | | |
| | CPLX3 | 0.90 | 2.15 | 0.99 | | | |
| **TMS** | TMS1 | 0.84 | 3.15 | 0.88 | 0.86 | 0.91 | 0.71 |
| | TMS2 | 0.90 | 3.28 | 0.82 | | | |
| | TMS3 | 0.84 | 3.23 | 0.85 | | | |
| | TMS4 | 0.79 | 3.18 | 0.91 | | | |
| **ITS** | ITS1 | 0.87 | 2.95 | 1.04 | 0.75 | 0.85 | 0.66 |
| | ITS2 | 0.80 | 2.88 | 0.99 | | | |
| | ITS3 | 0.77 | 2.75 | 1.11 | | | |
| **PEOU** | PEOU1 | 0.79 | 3.42 | 0.93 | 0.81 | 0.89 | 0.73 |
| | PEOU2 | 0.87 | 3.55 | 0.88 | | | |
| | PEOU3 | 0.90 | 3.60 | 0.85 | | | |
| **PU** | PU1 | 0.82 | 3.58 | 0.91 | 0.90 | 0.92 | 0.66 |
| | PU5 | 0.82 | 3.61 | 0.87 | | | |
| | PU6 | 0.82 | 3.59 | 0.88 | | | |
| | PU7 | 0.79 | 3.53 | 0.90 | | | |
| | PU8 | 0.78 | 3.50 | 0.92 | | | |
| | PU9 | 0.85 | 3.64 | 0.86 | | | |
| **BI** | BI1 | 0.89 | 3.79 | 0.82 | 0.86 | 0.91 | 0.78 |
| | BI2 | 0.88 | 3.75 | 0.85 | | | |
| | BI3 | 0.86 | 3.68 | 0.88 | | | |
| **CP** | CP1 | 0.96 | 3.12 | 1.05 | 0.91 | 0.96 | 0.91 |
| | CP2 | 0.96 | 3.10 | 1.08 | | | |

AVE: average variance extracted, BI: behavioral intention, COMP: compatibility, CP: competitive pressure, CPLX: complexity, CR: composite reliability, ITS: IT support and training, M: mean, PEOU: perceived ease of use, PU: perceived usefulness, RA: relative advantage, SD: standard deviation, SQ: system quality, TMS: top management support, α: Cronbach's α

all predictor variables. S4 Table shows variance inflation factor values ranging from 1.4 (e.g., ITS3, SQ2, SQ3) to 3.3 (TMS2), all well below the conservative threshold of 5, confirming that the predictors did not suffer from multicollinearity issues.

Assessing model fit is critical for determining how well the measurement model represents the underlying data and for establishing its credibility for generalization. The mean absolute differences between predicted and observed covariances

**Table 4. Discriminant validity (√average variance extracted on diagonal; inter-construct correlations off-diagonal).**

| Construct | RA | SQ | COMP | CPLX | TMS | ITS | PEOU | PU | BI | CP |
|---|---|---|---|---|---|---|---|---|---|---|
| RA | **0.82** | 0.59 | 0.75 | 0.21 | 0.75 | 0.61 | 0.67 | 0.69 | 0.58 | 0.63 |
| SQ | 0.58 | **0.8** | 0.59 | 0.15 | 0.61 | 0.68 | 0.66 | 0.76 | 0.58 | 0.59 |
| COMP | 0.75 | 0.59 | **0.88** | 0.25 | 0.71 | 0.57 | 0.69 | 0.64 | 0.62 | 0.64 |
| CPLX | 0.21 | 0.15 | 0.25 | **0.9** | 0.27 | 0.11 | 0.22 | 0.23 | 0.23 | 0.24 |
| TMS | 0.75 | 0.61 | 0.71 | 0.27 | **0.84** | 0.56 | 0.68 | 0.73 | 0.68 | 0.67 |
| ITS | 0.61 | 0.68 | 0.57 | 0.11 | 0.56 | **0.81** | 0.62 | 0.71 | 0.52 | 0.67 |
| PEOU | 0.67 | 0.66 | 0.69 | 0.22 | 0.68 | 0.62 | **0.85** | 0.8 | 0.74 | 0.74 |
| PU | 0.69 | 0.76 | 0.64 | 0.23 | 0.73 | 0.71 | 0.8 | **0.81** | 0.73 | 0.77 |
| BI | 0.58 | 0.58 | 0.62 | 0.23 | 0.68 | 0.52 | 0.74 | 0.73 | **0.88** | 0.83 |
| CP | 0.63 | 0.59 | 0.64 | 0.24 | 0.67 | 0.67 | 0.74 | 0.77 | 0.83 | **0.96** |

Diagonals (in bold) are the square roots of AVE for each construct. Off-diagonal cells are the Pearson correlations between constructs. Every diagonal element exceeds its corresponding row/column correlations, confirming discriminant validity via the Fornell-Larcker criterion. BI: behavioral intention, COMP: compatibility, CP: competitive pressure, CPLX: complexity, ITS: IT support and training, PEOU: perceived ease of use, PU: perceived usefulness, RA: relative advantage, SQ: system quality, TMS: top management support.

**Table 5. Heterotrait-Monotrait (HTMT) ratio matrix for discriminant validity.**

| | BI | PEOU | PU | RA | SQ | COMP | CPLX | ITS | TMS | CP |
|---|---|---|---|---|---|---|---|---|---|---|
| BI | – | 0.9 | 0.8 | 0.7 | 0.7 | 0.7 | 0.2 | 0.7 | 0.8 | 0.9 |
| PEOU | | – | 0.9 | 0.8 | 0.9 | 0.8 | 0.2 | 0.8 | 0.8 | 0.9 |
| PU | | | – | 0.8 | 0.9 | 0.7 | 0.2 | 0.8 | 0.8 | 0.9 |
| RA | | | | – | 0.7 | 0.9 | 0.2 | 0.7 | 0.9 | 0.7 |
| SQ | | | | | – | 0.7 | 0.1 | 0.9 | 0.8 | 0.7 |
| COMP | | | | | | – | 0.2 | 0.7 | 0.8 | 0.7 |
| CPLX | | | | | | | – | 0.1 | 0.3 | 0.2 |
| ITS | | | | | | | | – | 0.7 | 0.8 |
| TMS | | | | | | | | | – | 0.8 |
| CP | | | | | | | | | | – |

BI: behavioral intention, COMP: compatibility, CP: competitive pressure, CPLX: complexity, ITS: IT support and training, PEOU: perceived ease of use, PU: perceived usefulness, RA: relative advantage, SQ: system quality, TMS: top management support.

as assessed using SRMR values were used to indicate the model fit. S5 Table reports SRMR values of 0.075 for the saturated model and 0.079 for the estimated model, both under the 0.10 cut-off, indicating a good overall model fit.

## Measurement model refinement

During the measurement evaluation phase, validity and reliability concerns necessitated model refinements. In particular, indicators with outer loadings < 0.7, Cronbach's α values < 0.7, actual loading values < 0.7, and AVE values < 0.5 were removed to improve overall model criteria. This refinement led to the elimination of several low-performing items, ensuring that each construct retained at least three robust indicators. Post-refinement, composite reliability improved to range from 0.86 to 0.94, and AVE values increased to between 0.61 and 0.88, confirming that the revised constructs exceed psychometric thresholds. The refined model's indicators exhibited acceptable outer loading values, reinforcing the reliability and validity of the measurement model. Table 6 details the final indicator set with their standardized loadings, and Fig 2

**Table 6. Refined items of the final measurement model.**

| Construct | # | Item |
|---|---|---|
| RA | 1 | Adopting EHIS improves the quality of my work. |
| | 2 | Using EHIS increases my job performance. |
| | 3 | Using EHIS enables me to quickly and easily obtain information on investigation or treatment procedures. |
| | 4 | Using EHIS improves patient safety. |
| | 5 | Using EHIS reduces time and costs. |
| COMP | 1 | The current IT infrastructure supports the use of electronic EHIS. |
| | 2 | Using EHIS fits well with my work style. |
| | 3 | Using EHIS is fully compatible with my current situation. |
| CPLX | 4 | I believe that EHIS is complicated to use. |
| | 5 | It is difficult for me to remember how to perform tasks using EHIS. |
| | 6 | Adopting EHIS requires a significant amount of mental effort. |
| TMS | 1 | TMS is important for adopting EHIS. |
| | 2 | Support from related departments is important for adopting EHIS. |
| | 3 | Management provided helpful support during the implementation of EHIS. |
| | 4 | Management expects me to use EHIS. |
| ITS | 1 | I have received sufficient formal training to use EHIS. |
| | 2 | The IT staff provided adequate support for EHIS. |
| | 3 | I will use EHIS if I receive proper training. |
| SQ | 1 | The information provided by EHIS is always accurate. |
| | 2 | The information provided by EHIS is always timely. |
| | 3 | I find the EHIS interface to be user-friendly. |
| PU | 1 | I have access to the information where I need it. |
| | 2 | The information provided by EHIS is always updated. |
| | 3 | The data I record are important for patient care. |
| | 4 | I am confident in the reliability of the documented data. |
| | 5 | Using EHIS avoids duplication of examinations. |
| | 6 | Using EHIS reduces the risk of errors. |
| PEOU | 1 | It is easy to learn how to use EHIS. |
| | 2 | It is easy to use EHIS. |
| | 3 | It is easy to understand how to perform the intended tasks using EHIS. |
| BI | 1 | When available in my clinical practice, I intend to use EHIS for all my clinical activities. |
| | 2 | When available in my community, I intend to adopt EHIS for all my clinical activities. |
| | 3 | The likelihood that I will use EHIS for all my clinical activities, when available in my organization, is very high. |
| CP | 1 | We are aware of EHIS implementation in our hospital. |
| | 2 | We understand the competitive advantages offered by EHIS in our hospital. |

BI: behavioral intention, COMP: compatibility, CP: competitive pressure, CPLX: complexity, ITS: IT support and training, PEOU: perceived ease of use, PU: perceived usefulness, RA: relative advantage, SQ: system quality, TMS: top management support.

depicts the refined measurement model. As depicted in Fig 2, the structural model explains 68% of the variance in PU, 57% in PEOU, and 74% in BI, underscoring the model's strong predictive power.

## Structural model findings

**Path analysis and hypothesis testing.** To assess the strength of the relationships between constructs, we examined both the path coefficients and their associated p-values. In our analysis, a path coefficient of 0.1 or above was considered

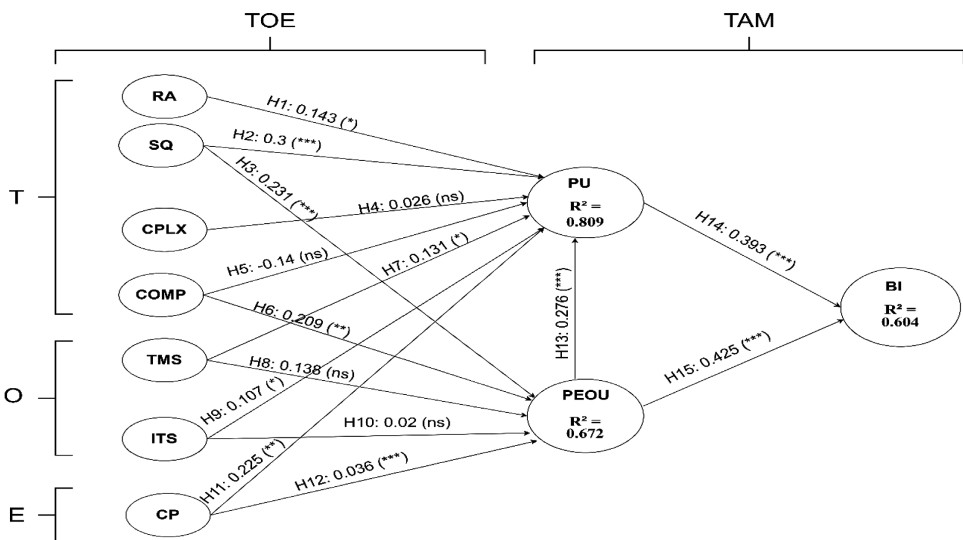

**Fig 2. Factor relationships in the structural model.** BI: behavioral intention, COMP: compatibility, CP: competitive pressure, CPLX: complexity, ITS: IT support and training, PEOU: perceived ease of use, PU: perceived usefulness, RA: relative advantage, SQ: system quality, TMS: top management support. p: p-value, *p < 0.05. **p < 0.01. ***p < 0.001. ns: not significant.

indicative of a meaningful relationship, while p-values below 0.05 confirm statistical significance. The hypothesis testing results reveal a robust and statistically significant network of relationships among the constructs. Ordered by the magnitude of the path coefficients, the strongest effect is observed for the relationship between PEOU and BI ($\beta = 0.43$, $p < 0.001$), which highlights that system usability is a critical factor associated with users' intentions. Following closely, PU was associated with BI ($\beta = 0.39$, $p < 0.001$), underscoring the notion that both ease and utility are essential for promoting system adoption. In addition, SQ has a significant positive influence on PU ($\beta = 0.30$, $p < 0.001$), while PEOU contributes to PU ($\beta = 0.28$, $p < 0.001$), indicating that not only does the SQ matter, but its PEOU further enhances its PU. CP appears to reinforce these effects by significantly impacting both PEOU ($\beta = 0.23$, $p < 0.001$) and PU ($\beta = 0.23$, $p = 0.004$). Further, SQ and COMP also play a meaningful role in shaping PEOU, with coefficients of 0.23 ($p = 0.001$) and 0.21 ($p = 0.006$), respectively. Although the effects of RA ($\beta = 0.14$, $p = 0.012$), TMS ($\beta = 0.13$, $p = 0.049$), and ITS ($\beta = 0.048$, $p = 0.048$) on PU are relatively smaller, they are nonetheless significant and contribute to the overall understanding of the factors influencing system acceptance. Collectively, these findings underscore the importance of usability, usefulness, and supportive external factors in driving the adoption of electronic health information systems among healthcare providers. Table 7 details these key values for the hypothesized paths.

### Hierarchical regression analyses results

To assess the effects of our control variables (training, job title, and years of experience) from those of our theoretical predictors, we conducted a two-step hierarchical regression for each composite outcome. Table 8 presents the first step (controls only) and Table 9 the second step (adding PEOU, PU, BI, CP, ITS, TMS, SQ, COMP, and RA). Step 1 assessed how much variance was explained solely by the control variables and step 2 then tested the incremental contribution of the nine theory-driven constructs over and above those controls.

In the first step of the hierarchical analyses (Table 8), the three control variables, training, job title (role), and years of experience, together explained between 1.9% and 8.8% of variance across the ten composite outcomes ($\Delta R^2 = 0.019$

**Table 7. Path coefficients and hypothesis testing.**

| Hypothesis | Construct A→B | Path coefficient (β) | p | Supported |
|---|---|---|---|---|
| H1 | RA→PU | 0.14 | **0.012** | Yes |
| H2 | SQ→PU | 0.30 | **< 0.001** | Yes |
| H3 | SQ→PEOU | 0.23 | **0.001** | Yes |
| H4 | CPLX→PU | 0.03 | 0.460 | No |
| H5 | COMP→PU | -0.14 | 0.050 | No |
| H6 | COMP→PEOU | 0.21 | **0.006** | Yes |
| H7 | TMS→PU | 0.13 | **0.049** | Yes |
| H8 | TMS→PEOU | 0.14 | 0.130 | No |
| H9 | ITS→PU | 0.11 | **0.048** | Yes |
| H10 | ITS→PEOU | 0.02 | 0.770 | No |
| H11 | CP→PU | 0.23 | **0.004** | Yes |
| H12 | CP→PEOU | 0.37 | **< 0.001** | Yes |
| H13 | PEOU→PU | 0.28 | **< 0.001** | Yes |
| H14 | PU→BI | 0.39 | **< 0.001** | Yes |
| H15 | PEOU→BI | 0.43 | **< 0.001** | Yes |

BI: behavioral intention, COMP: compatibility, CP: competitive pressure, CPLX: complexity, ITS: IT support and training, p: p-value, PEOU: perceived ease of use, PU: perceived usefulness, RA: relative advantage, SQ: system quality, TMS: top management support. Statistically significant p-values are in boldface.

**Table 8. Effects of control variables on composite outcomes (step 1).**

| Outcome | ΔR² | F-change | p | Training β | Training p | Job title β | Job title p | Experience β | Experience p |
|---|---|---|---|---|---|---|---|---|---|
| RA | 0.04 | 2.96 | **0.033** | 0.19 | **0.005** | 0.01 | 0.930 | -0.08 | 0.230 |
| COMP | 0.02 | 1.52 | 0.210 | 0.09 | 0.190 | 0.10 | 0.120 | 0.05 | 0.480 |
| CPLX | 0.07 | 5.57 | **0.001** | 0.04 | 0.600 | -0.16 | **0.020** | 0.20 | **0.003** |
| TMS | 0.05 | 3.49 | **0.017** | 0.19 | **0.004** | -0.09 | 0.190 | -0.02 | 0.810 |
| ITS | 0.09 | 6.93 | **< 0.001** | 0.28 | **< 0.001** | 0.06 | 0.380 | 0.06 | 0.400 |
| SQ | 0.09 | 6.67 | **< 0.001** | 0.29 | **< 0.001** | 0.05 | 0.420 | 0.00 | 0.960 |
| PU | 0.06 | 4.43 | **0.005** | 0.22 | **< 0.001** | -0.07 | 0.280 | -0.10 | 0.130 |
| PEOU | 0.02 | 1.41 | 0.240 | 0.13 | 0.060 | 0.04 | 0.560 | 0.03 | 0.650 |
| BI | 0.03 | 1.84 | 0.140 | 0.14 | **0.036** | -0.02 | 0.770 | 0.04 | 0.520 |
| CP | 0.03 | 2.30 | 0.080 | 0.18 | **0.011** | -0.02 | 0.740 | -0.01 | 0.870 |

BI: behavioral intention, COMP: compatibility, CP: competitive pressure, CPLX: complexity, ITS: IT support and training, p: p-value, PEOU: perceived ease of use, PU: perceived usefulness, RA: relative advantage, SQ: system quality, TMS: top management support. Statistically significant p-values are in boldface.

to 0.088). Significant incremental effects emerged for RA ($\Delta R^2 = 0.040$, $p = 0.033$), CPLX ($\Delta R^2 = 0.072$, $p = 0.001$), TMS ($\Delta R^2 = 0.046$, $p = 0.017$), ITS ($\Delta R^2 = 0.088$, $p < 0.001$), SQ ($\Delta R^2 = 0.085$, $p < 0.001$), and PU ($\Delta R^2 = 0.058$, $p = 0.005$), whereas COMP, PEOU, BI, and CR did not reach significance at this stage. Training was positively and significantly associated with RA ($\beta = 0.192$, $p = 0.005$), ITS ($\beta = 0.282$, $p < 0.001$), SQ ($\beta = 0.290$, $p < 0.001$), PU ($\beta = 0.219$, $p < 0.001$), and CR ($\beta = 0.175$, $p = 0.011$). Clinician status was negatively associated with CPLX ($\beta = -0.155$, $p < 0.020$), and years of experience was a significant positive associated factor with CPLX ($\beta = 0.199$, $p < 0.003$). No other control effects were significant in this step.

**Table 9. Incremental effects of theoretical associated factors (step 2).**

| Outcome | ΔR² | F-change | p | Top associated factors (β; P) |
|---|---|---|---|---|
| RA | 0.666 | 52.056 | **< 0.001** | COMP 0.395 (**< 0.001**); TMS 0.336 (**< 0.001**); ITS 0.190 (**0.001**) |
| COMP | 0.661 | 47.846 | **< 0.001** | RA 0.427 (**< 0.001**); TMS 0.214 (.003); SQ 0.209 (0.**002**) |
| CPLX | 0.079 | 2.154 | **0.027** | TMS -0.284 (**0.014**) |
| TMS | 0.653 | 49.954 | **< 0.001** | RA 0.343 (**< 0.001**); COMP 0.202 (0.**003**); BI 0.217 (0.**004**) |
| ITS | 0.508 | 28.957 | **< 0.001** | SQ 0.327 (**< 0.001**); PU 0.276 (0.**007**); RA 0.261 (**0.001**) |
| SQ | 0.569 | 37.871 | **< 0.001** | PU 0.499 (**< 0.001**); ITS 0.280 (**< 0.001**); COMP 0.227 (**0.002**) |
| PU | 0.759 | 95.192 | **< 0.001** | PEOU 0.294 (**< 0.001**); SQ 0.265 (**< 0.001**); CP 0.188 (**0.002**) |
| PEOU | 0.718 | 62.973 | **< 0.001** | PU 0.420 (**<.001**); RA 0.148 (.**023**); BI 0.172 (.**014**) |
| BI | 0.719 | 64.738 | **< 0.001** | CP 0.569 (**< 0.001**); TMS 0.185 (**0.004**); PEOU 0.168 (**0.014**) |
| CP | 0.738 | 73.598 | **< 0.001** | BI 0.514 (**< 0.001**); PU 0.237 (**0.002**); ITS 0.142 (**0.006**) |

BI: behavioral intention, COMP: compatibility, CP: competitive pressure, CPLX: complexity, ITS: IT support and training, p: p-value, PEOU: perceived ease of use, PU: perceived usefulness, RA: relative advantage, SQ: system quality, TMS: top management support. Statistically significant p-values are in boldface.

In the second step of each hierarchical regression (Table 9), adding the nine theoretical predictors (PEOU, PU, BI, CP, ITS, TMS, SQ, COMP, RA) led to a dramatic improvement in model fit across all ten composite outcomes. ΔR² ranged from 0.508 (for ITS) to 0.759 (for PU), with all F-change statistics highly significant ($p < 0.001$), indicating these constructs explain an additional 50.8 to 75.9% of variance over and above controls. Examining standardized betas, task-technology fit variables (COMP, TMS, ITS, and SQ) and both PU and PEOU emerged as the strongest drivers, though their relative importance varied by outcome. For instance, performance outcomes such as RA were chiefly associated with COMP ($β = 0.395$, $p < 0.001$) and TMS ($β = 0.336$, $p < 0.001$), whereas BI was dominated by CPLX ($β = 0.569$, $p < 0.001$) and PU ($β = 0.168$, $p = 0.014$). Perceptual constructs themselves exhibited the expected patterns, PEOU was strongly associated with PU ($β = 0.294$, $p < 0.001$) and PU was a key factor associated with SQ ($β = 0.499$, $p < 0.001$) and CP ($β = 0.237$, $p < 0.001$). Overall, the full models demonstrate that, beyond demographic controls, technology-related beliefs and usage constructs account for the largest share of explained variance in each composite score.

## Discussion

### Principal findings

Our extended TAM-TOE instrument demonstrated strong psychometric performance across all measurement criteria. Outer loadings for retained indicators ranged from 0.76 to 0.96, surpassing the 0.70 threshold for indicator reliability. Internal consistency was confirmed by Cronbach's α values between 0.72 and 0.91 and composite reliability from 0.84 to 0.96. Convergent validity was supported by AVE scores of 0.64 to 0.91, and discriminant validity was established via both the Fornell-Larcker criterion and HTMT ratios below 0.90, in line with best practices for PLS-SEM applications [47,49,51,52].

The structural model explained substantial variance in core TAM constructs: 68% of PU, 57% of PEOU, and 74% of BI. These R² values exceed conventional benchmarks (≥ 0.20) and compare favorably to prior studies, which often report moderate explanatory power when applying TAM in healthcare settings [4,5,9,12,14,15,21-24,31,32,39]. Hierarchical regression further showed that our theoretical predictors contributed an additional 50.8% to 75.9% of variance beyond demographic controls, underscoring the robust incremental validity of the extended model. These values are notably higher than those typically reported in recent TAM/TOE studies conducted in low- and middle-income countries, which often range between 40–60% for PU and BI [20]. This underscores the strong explanatory power of our integrated TAM–TOE paradigm in resource-limited hospital settings.

Consistent with Davis's original TAM, PEOU (β = 0.43, p < 0.001) and PU (β = 0.39, p < 0.001) emerged as the strongest factor associated with BI, reaffirming that ease of use and perceived value remain key levers of technology adoption in low-resourced hospitals [12,22]. SQ significantly enhanced both PU (β = 0.30, p < 0.001) and PEOU (β = 0.23, P = 0.001), highlighting the central role of stable, user-friendly system design in shaping user perceptions [16].

Environmental pressure (CP) exerted medium-to-large effects on both PEOU (β = 0.37, p < 0.001) and PU (β = 0.23, p = 0.004), indicating that awareness of peer-institution implementations and competitive dynamics strongly motivate acceptance in settings where resources are constrained [11,41]. COMP also bolstered PEOU (β = 0.21, p = 0.006), suggesting that alignment with existing workflows reduces the perceived effort required to use EHIS [5].

Organizational supports showed more nuanced effects. TMS and ITS significantly influenced PU (β = 0.13, p = 0.049; β = 0.11, p = 0.048, respectively) but did not significantly affect PEOU. This diverges from some studies where formal training directly eased system use [18,24], implying that in our context, users' judgments of usefulness are more sensitive to leadership endorsement and technical assistance than are their ease-of-use perceptions.

The non-significant effect of CPLX on PU (β = 0.03, p = 0.460) suggests that, in this sample, system complexity did not substantially influence perceptions of usefulness. These results were not consistent with reports of complexity as a barrier in other low-resource environments [8,10]. This non-effect may reflect effective user adaptation or streamlined system interfaces that mitigate perceived difficulty. This may be explained by the participants' extensive experience with computers and EHISs (mean = 10.4 years), which likely mitigated the perceived impact of complexity.

Collectively, these principal findings affirm that core TAM relationships hold in resource-limited contexts while underscoring the amplified importance of system quality and environmental drivers. They also reveal context-specific dynamics in the roles of organizational support and complexity, offering new insights for tailoring EHIS implementation strategies.

While most hypothesized relationships were supported, several structural paths yielded counterintuitive or non-significant results that merit closer examination. These findings highlight contextual dynamics unique to Palestinian hospitals and contribute to the broader discourse on technology acceptance in resource-constrained settings. For instance, the negative and borderline significant path from COMP to PU (β = -0.14, p = 0.050) challenges conventional TAM/TOE expectations. Typically, systems that align with existing workflows are perceived as more useful. However, in our context, high compatibility may inadvertently reinforce entrenched paper-based or inefficient practices. When EHIS replicate legacy processes without introducing transformative efficiencies, providers may view them as automating inefficiency rather than delivering meaningful improvements. Similar patterns have been reported in low- and middle-income countries, where "too compatible" systems fail to disrupt outdated routines and thus diminish perceived usefulness [53-54]. This suggests that in resource-limited environments, compatibility must be balanced with innovation, systems should respect existing workflows but also introduce visible enhancements that signal genuine progress [55,54]. Moreover, TMS→PEOU was non-significant, despite TMS significantly influencing PU. This indicates that leadership commitment and resource allocation help users appreciate the strategic value of EHIS but do not necessarily reduce perceptions of effort. In practice, managerial support often manifests as policy directives or resource provision rather than hands-on facilitation [53,55]. Without direct involvement in simplifying interfaces or streamlining training, users may continue to perceive the system as effortful, even if they recognize its organizational importance. Furthermore, ITS→PEOU was strongly non-significant, though ITS significantly enhanced PU. This pattern suggests that current training modalities and IT support structures in Palestinian hospitals emphasize the utility of EHIS (e.g., demonstrating reporting functions or compliance benefits) but do not adequately reduce operational difficulty. Didactic training sessions, reactive help-desk models, and limited opportunities for experiential learning may fail to address the day-to-day challenges of system navigation [56,57]. Consequently, users understand the system's value but still perceive it as cumbersome to operate. This underscores the need for more interactive, hands-on training and proactive IT support that directly target ease of use. Taken together, these findings reveal that contextual factors can invert or mute expected TAM/TOE relationships. They highlight the importance of designing EHIS interventions that not only demonstrate strategic value but also tangibly

reduce effort and disrupt inefficient legacy workflows. Addressing these dynamics may be critical for accelerating EHIS adoption in resource-constrained health systems.

It is important to note that the CPLX→PEOU path, commonly reported in TAM studies, was omitted from our final model following pilot testing. Although theoretically relevant, the path demonstrated poor psychometric performance in our context, with unstable loadings and weak explanatory contribution [50]. This suggests that in Palestinian hospitals, perceptions of system complexity may not directly translate into judgments of ease of use, possibly because users already anticipate EHIS to be effortful regardless of technical design. Instead, complexity appears to influence perceived usefulness more strongly, reflecting concerns about whether difficult systems can deliver tangible benefits. Future studies may reintroduce this path with refined measurement instruments to further explore its role in resource-limited settings.

It is also noteworthy that the CPLX→PU path was non-significant. In mandatory-use environments such as Palestinian hospitals, providers may have limited discretion to reject EHIS regardless of perceived difficulty. As a result, effort expectancy becomes less influential in shaping usefulness judgments, since clinicians must engage with the system even if it is complex. This dynamic may explain why complexity was not significantly associated with PU in our model.

It is noteworthy that CP exerted unusually strong effects on both PU ($\beta = 0.23$, $p = 0.004$) and PEOU ($\beta = 0.37$, $p < 0.001$). These coefficients are among the highest reported in healthcare TAM/TOE studies, underscoring the pivotal role of external institutional forces in shaping technology acceptance in low- and middle-income contexts [20]. In environments where hospitals face mounting pressure to modernize and demonstrate efficiency, competitive dynamics appear to accelerate both perceptions of usefulness and ease of use, amplifying adoption beyond what is typically observed in voluntary-use settings.

## Implications of the findings

Our findings offer several actionable insights for healthcare administrators, technology implementers, and researchers working in resource-limited settings.

Hospitals should prioritize system quality and usability by investing in stable, intuitive EHIS interfaces and conducting iterative usability testing to catch design flaws early, since high system quality was associated with both PU and PEOU [16]. They can also leverage environmental pressure, awareness of peer institutions' EHIS adoption proved a powerful motivator, by working with ministries of health and professional associations to establish communities of practice or benchmarking programs that spotlight success stories and encourage wider uptake [41]. Aligning EHIS with existing workflows is equally critical: implementation teams ought to map clinical processes before deployment and configure system modules to mirror familiar routines, thereby reducing the learning curve and enhancing PEOU [5]. Finally, although TMS and ITS improve PU, they do not automatically translate into seamless system operation; training curricula should therefore extend beyond one-off sessions to include on-the-job coaching and peer mentoring, turning leadership endorsement into real user confidence [18,22].

Our robust psychometric validation of the combined TAM-TOE instrument indicates that it can be deployed across diverse low- and middle-income country settings to benchmark and compare the key drivers of EHIS acceptance [14]. Although our cross-sectional analysis showed that perceived system complexity did not significantly undermine usefulness, future longitudinal or mixed-methods studies should investigate how perceptions of complexity evolve with continued use and user familiarity [10]. Moreover, expanding the model to include additional moderators and mediators, such as technology self-efficacy, resource availability, and cultural norms, will refine our understanding of the nuanced, context-specific factors shaping behavioral intention toward EHIS adoption.

Policymakers and regulatory bodies should embed pre-implementation EHIS readiness assessments into accreditation and funding criteria to proactively identify and address organizational or environmental barriers before large-scale roll-outs [31]. They must also allocate dedicated budget lines for ongoing technical assistance, refresher training, and maintenance support rather than one-time expenditures, ensuring that ITS translates into sustained system use [8]. Furthermore,

promoting regional collaboration through coordinated EHIS communities of practice will enable governmental and nongovernmental organizations to pool lessons learned, share resources, and harness competitive pressure as a positive factor associated with innovation [39].

The strong influence of competitive pressure suggests that hospitals in Palestine are motivated not only by internal efficiency but also by the need to demonstrate digital readiness relative to peer institutions. In practice, this manifests in efforts to modernize EHIS platforms to attract patients, secure funding, and maintain reputational standing. Policymakers should recognize these dynamics and leverage competitive benchmarking as a factor associated with adoption, while ensuring that competition does not exacerbate inequities between better-resourced and resource-limited hospitals.

By translating our empirical insights into these concrete strategies, stakeholders can more effectively influence EHIS adoption, optimize resource utilization, and ultimately improve patient care outcomes in challenging healthcare environments.

## Limitations and future directions

Our study has several limitations that warrant consideration. First, the cross-sectional survey design limits our ability to draw causal inferences among TAM-TOE constructs and behavioral intention. Second, data were collected through an interviewer-administered questionnaires, which may introduce common-method variance and social desirability bias. Third, our sample comprised staff from a select group of hospitals in resource-constrained settings, potentially restricting the generalizability of findings to other facility types or regions. Fourth, although the response rate was high (88%), the use of purposive sampling and the possibility of non-response bias should be acknowledged. Those who declined participation may differ in their perceptions of EHIS, which could limit the generalizability of findings. Fifth, although participants were recruited from a wide range of hospital departments and units, including emergency, outpatient clinics, inpatient wards, laboratory, radiology, and pharmacy, the analysis did not explicitly compare responses across these departmental categories. Our statistical comparisons focused on clinical versus nonclinical roles rather than unit-level differences. As workflows and system usage patterns may vary substantially between departments, future studies should examine these variations more closely to provide deeper insights into EHIS acceptance across different hospital contexts. Sixth, we measured intention rather than actual system use or patient-level outcomes, leaving the connection between EHIS acceptance and clinical impact unexplored. Another limitation concerns the complexity of the proposed research model. With seven exogenous predictors and 15 hypothesized paths, there is potential theoretical overlap between constructs such as RA versus SQ, or COMP versus PEOU. While discriminant validity was established statistically, the inclusion of multiple predictors may dilute the explanatory power of individual factors or obscure the most critical drivers of EHIS adoption. Future research should consider more parsimonious models that balance comprehensiveness with theoretical clarity, thereby isolating the most influential determinants of acceptance in resource-limited hospital contexts.

Looking ahead, future research should adopt longitudinal or experimental designs to track changes in perceptions and behavior over time and establish causal pathways between determinants and EHIS uptake. Embedding qualitative methods, such as interviews and focus groups, will uncover deeper organizational and cultural insights that quantitative surveys may overlook. Researchers could enrich the TAM-TOE framework by integrating additional constructs like technology self-efficacy, leadership commitment, and resource constraints [41]. Crucially, studies linking EHIS acceptance to objective metrics of workflow efficiency, data quality, and patient outcomes will strengthen the evidence base for policymakers and implementers. Finally, rigorous cost-benefit analyses and multi-site implementation trials across diverse healthcare contexts will help tailor deployment strategies to local needs and resource realities.

## Conclusion

Our study validated a combined TAM-TOE instrument with strong reliability and model fit for assessing electronic health information system acceptance in resource-limited hospitals. Structural equation modeling showed that SQ, COMP with

existing workflows, environmental pressure from peer institutions (CP), TMS, and targeted ITS significantly were associated with PU, PEOU, and ultimately BI to adopt EHIS. By pinpointing these key determinants, our findings offer a clear roadmap for implementers and policymakers: prioritize stable, intuitive system design; tailor deployments to mirror clinical routines; foster supportive leadership and learning networks; and leverage peer success stories to build momentum. Applying our validated assessment tool before and during roll-out can guide resource allocation, refine training programs, and monitor adoption trajectories. Ultimately, these evidence-based strategies can accelerate EHIS uptake, enhance data-driven decision-making, and improve patient care outcomes in challenging healthcare environments.

## Supporting information

**S1 Table. Adherence to the STROBE statement.**
(DOCX)

**S2 Table. Operationalization of constructs and indicators for EHIS acceptance and satisfaction – developed and adapted items.**
(DOCX)

**S3 Table. The study questionnaire.**
(DOCX)

**S1 Fig. Flow diagram of participant recruitment.**
(TIFF)

**S4 Table. Variance inflation factor values.**
(DOCX)

**S5 Table. Standardized root mean square residual.**
(DOCX)

## Acknowledgments

An-Najah National University (www.najah.edu) and An-Najah National University Hospital (www.nnuh.org) are acknowledged for making this study possible. The authors would like to thank also the participating hospitals for allowing this study. This work is based on the master thesis of Ahmad Dweikat.

## Author contributions

**Conceptualization:** Ramzi Shawahna, Nidal Yousef Dwaikat.

**Data curation:** Ahmad Dweikat.

**Formal analysis:** Ramzi Shawahna, Ahmad Dweikat.

**Investigation:** Ramzi Shawahna, Ahmad Dweikat.

**Methodology:** Ramzi Shawahna, Ahmad Dweikat, Nidal Yousef Dwaikat.

**Project administration:** Ramzi Shawahna.

**Software:** Ramzi Shawahna.

**Supervision:** Ramzi Shawahna, Nidal Yousef Dwaikat.

**Validation:** Ramzi Shawahna, Ahmad Dweikat.

**Visualization:** Ramzi Shawahna, Ahmad Dweikat, Nidal Yousef Dwaikat.

**Writing – original draft:** Ramzi Shawahna, Ahmad Dweikat, Nidal Yousef Dwaikat.

**Writing – review & editing:** Ramzi Shawahna, Ahmad Dweikat, Nidal Yousef Dwaikat.

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
