## [Decision Letter · Decision Letter 0]

12 Dec 2025

Response to Reviewers'. This file does not need to include responses to any formatting updates and technical items listed in the 'Journal Requirements' section below.'. This file does not need to include responses to any formatting updates and technical items listed in the 'Journal Requirements' section below.* A marked-up copy of your manuscript that highlights changes made to the original version. You should upload this as a separate file labeled 'Revised Manuscript with Track Changes'.'.* An unmarked version of your revised paper without tracked changes. You should upload this as a separate file labeled 'Manuscript'.'. If you would like to make changes to your financial disclosure, competing interests statement, or data availability statement, please make these updates within the submission form at the time of resubmission. Guidelines for resubmitting your figure files are available below the reviewer comments at the end of this letter. We look forward to receiving your revised manuscript. Kind regards, Phat Huynh

**Journal Requirements:**

1. In the online submission form, you indicated that The data used in this study are provided in the manuscript and supplementary materials. The datasets used in the analysis or entered into statistical software can be obtained from the corresponding author upon making a reasonable request.

3. Uploaded as supplementary information.

2. Please provide an Author Summary. This should appear in your manuscript between the Abstract (if applicable) and the Introduction, and should be 150–200 words long. The aim should be to make your findings accessible to a wide audience that includes both scientists and non-scientists. Sample summaries can be found on our website under Submission Guidelines:

https://journals.plos.org/digitalhealth/s/submission-guidelines#loc-parts-of-a-submission

3. Please provide separate figure files in .tif or .eps format.

4. We have noticed that you have uploaded Supporting Information files, but you have not included a list of legends. Please add a full list of legends for your Supporting Information files after the references list.

**Reviewers' Comments:** Reviewer's Responses to Questions

**Comments to the Author**

1. Does this manuscript meet PLOS Digital Health’s publication criteria? Is the manuscript technically sound, and do the data support the conclusions? The manuscript must describe methodologically and ethically rigorous research with conclusions that are appropriately drawn based on the data presented.? Is the manuscript technically sound, and do the data support the conclusions? The manuscript must describe methodologically and ethically rigorous research with conclusions that are appropriately drawn based on the data presented.

Reviewer #1: Yes

Reviewer #2: Yes

Reviewer #3: Partly

Reviewer #4: Yes

2. Has the statistical analysis been performed appropriately and rigorously?

Reviewer #1: Yes

Reviewer #2: I don't know

Reviewer #3: Yes

Reviewer #4: Yes

3. Have the authors made all data underlying the findings in their manuscript fully available (please refer to the Data Availability Statement at the start of the manuscript PDF file)?

The PLOS Data policy requires authors to make all data underlying the findings described in their manuscript fully available without restriction, with rare exception. The data should be provided as part of the manuscript or its supporting information, or deposited to a public repository. For example, in addition to summary statistics, the data points behind means, medians and variance measures should be available. If there are restrictions on publicly sharing data—e.g. participant privacy or use of data from a third party—those must be specified.requires authors to make all data underlying the findings described in their manuscript fully available without restriction, with rare exception. The data should be provided as part of the manuscript or its supporting information, or deposited to a public repository. For example, in addition to summary statistics, the data points behind means, medians and variance measures should be available. If there are restrictions on publicly sharing data—e.g. participant privacy or use of data from a third party—those must be specified.

Reviewer #1: No

Reviewer #2: No

Reviewer #3: Yes

Reviewer #4: Yes

4. Is the manuscript presented in an intelligible fashion and written in standard English?

Reviewer #1: Yes

Reviewer #2: Yes

Reviewer #3: Yes

Reviewer #4: Yes

Reviewer #1: Overall Assessment

This is a strong and well-structured manuscript that provides genuine theoretical and empirical value. The integration of the Technology Acceptance Model (TAM) with a carefully selected set of Technology–Organization–Environment (TOE) constructs is well justified and substantially improves explanatory power compared to TAM alone. The PLS-SEM analysis is executed rigorously, the psychometric properties are excellent, and the R² values especially 74% for Behavioral Intention are considerably higher than typical TAM-only healthcare studies. The contextual focus on Palestinian governmental and nongovernmental hospitals adds important relevance for digital health adoption in resource constrained environments.

However, several unexpected or contradictory structural paths are under-discussed or omitted entirely in the Discussion, which currently limits the paper’s interpretive depth and theoretical contribution. These findings are not routine non-supports; they represent potentially important mechanisms in resource-limited digital health settings and deserve explicit examination. Addressing these points will significantly enhance the manuscript’s impact. Therefore, I recommend Major Revision.

Major Compulsory Revisions

1. Unexpected or Negative Path Coefficients Need Dedicated Discussion

Several structural paths are surprising, counterintuitive, or theoretically meaningful but receive little or no discussion:

Compatibility → PU: β = –0.14, p = .05 (borderline significant negative)

TMS → PEOU: non-significant

ITS → PEOU: strongly non-significant

These results challenge conventional TAM/TOE relationships and may reflect important contextual dynamics in Palestinian health systems. For example:

The negative Compatibility → PU path may indicate that systems closely matching existing (often paper-based or inefficient) workflows are perceived as less useful, as they automate inefficiency rather than transform it. Some LMIC studies show similar patterns when “too compatible” systems fail to deliver real improvements.

The fact that TMS and ITS significantly influence PU but not PEOU suggests current training/support helps users understand the system’s value but does not reduce perceived effort common when training is didactic rather than hands-on, or when IT support is reactive.

Action required: Add a subsection (approx. 300–400 words) in the Discussion explicitly interpreting these findings. These insights could be among the most valuable contributions of the paper.

2. Missing Path: Complexity → PEOU

The model includes Complexity → PU but omits the standard Complexity → PEOU path widely reported in TAM literature.

Action required:

Either (1) justify the omission explicitly, or (2) test and report the CPLX → PEOU path post-hoc (even if non-significant). This will reinforce confidence in the model specification.

3. Data Availability Statement Not Compliant With PLOS Requirements

The current statement (“data available upon reasonable request”) does not comply with PLOS Digital Health’s mandatory open data policy.

Action required:

Deposit the anonymized dataset and/or the indicator correlations, means/SDs, and PLS-SEM bootstrapping output in a public repository (e.g., Zenodo, Figshare, Dryad) and update the Data Availability Statement accordingly.

4. Missing Structural Model Diagram (Figure 1)

Figure 1 is referenced several times but absent from the PDF.

Action required:

Include the full structural model diagram with standardized coefficients, significance markers, R² values, and f² where applicable.

5. Add a Summary Table of Hypothesis Testing

Results are currently dispersed across text.

Action required:

Add a table summarizing all hypotheses, including:

hypothesis number, path, β, t-value, p-value, f², and supported/not supported.

This is standard in PLS-SEM publications and improves readability.

6. Clarify Sampling Details and Response Rate

The Methods state that 220 healthcare professionals participated, but total questionnaires distributed, recruitment method, and response rate are not reported.

Action required:

Add 1–2 sentences clarifying:

number distributed

response rate

sampling method (convenience, purposive, stratified, etc.)

any potential non-response bias

This enhances transparency.

7. Strengthen Sample Size Justification

The manuscript currently foregrounds Raosoft (designed for simple random sampling), which is not ideal for PLS-SEM.

Action required:

Lead with the PLS-SEM-relevant justifications (inverse square root method, 10-times rule, or PLSpredict).

Raosoft may be retained as secondary or footnoted.

Minor Compulsory Revisions

Use “ranged from X to Y” rather than “ranged X–Y” when reporting indicator loadings.

Refine minor phrasing for clarity (e.g., “are more likely to perceive it as useful”).

Reduce repetition of “resource-limited” by occasionally using alternatives (“low-resource,” “constrained,” etc.).

Double-check that all β values in the abstract match those reported in the Results section.

Optional but Valuable Suggestions

Link the non-significance of Complexity → PU to mandatory-use environments where effort expectancy is less influential.

Add a brief comparison of your R² values with recent LMIC TAM/TOE studies to underscore the strength of your model.

Highlight the unusually strong competitive pressure effects (some of the highest reported in healthcare TAM literature).

Summary

This is a promising and well-constructed manuscript with strong methodological rigor, an excellent theoretical foundation, and important contextual relevance. The unexpected findings particularly the negative Compatibility → PU path and the selective influence of TMS/ITS are potentially the most policy-relevant contributions and deserve fuller exploration. Addressing the points above will significantly strengthen the paper’s theoretical clarity and practical value. I look forward to reviewing a revised version.

Reviewer #2: Pro 1: Strong methodological validation

The reviewer notes that one of the major strengths of this study is the thorough validation of the TAM-TOE instrument. The authors apply multiple validation methods—including indicator loadings, Cronbach’s α, composite reliability, AVE, Fornell–Larcker, HTMT, VIF checks, SRMR, and hierarchical regression—to support the reliability and validity of the constructs. The refinement steps are described clearly, and the hierarchical regression analysis demonstrates additional explanatory value beyond demographic variables. The reviewer encourages the authors to keep this level of methodological detail, as it supports the credibility of the findings.

Pro 2: Clear contextual grounding and focused use of extended TOE factors

The reviewer finds that the authors present an original and context-specific application of the TAM-TOE framework by situating the analysis within Palestinian hospitals facing resource limitations. The authors successfully relate technological and organizational factors—such as competitive pressure, system quality, and compatibility—to the realities of EHIS use in these settings. The reviewer appreciates how the discussion links the statistical results back to practical issues in this context, which strengthens the relevance of the study for local decision-makers and implementers.

Con 1: Questionnaire items should be introduced earlier and linked directly to item codes used in the Results

The reviewer suggests that the authors introduce questionnaire items earlier in the paper. In the current version, item codes such as RA1, CPLX2, and COMP3 appear in the Results section before readers know what the items actually measure. Because the wording of these items only appears later in the appendix or in subsequent tables, it becomes difficult for readers to interpret loadings and reliability values. The reviewer recommends adding a summary table of constructs and items in the Methods section or briefly listing item descriptions when they first appear in the Results.

Con 2: Results section would benefit from clearer organization regarding model refinement

The reviewer suggests reorganizing the Results section for better clarity. Table 2 appears early and seems to contain final results, but the narrative later introduces a refinement stage where several items are removed. This can cause confusion about which results correspond to the initial versus the refined model. The reviewer recommends:

explaining the refinement process before presenting the final measurement tables, or

explicitly labeling tables as “initial model” and “refined model.”

This restructuring would help readers follow the sequence of measurement evaluation and refinement more easily.

Reviewer #3: This manuscript presents an important and timely investigation of factors influencing electronic health information system (EHIS) acceptance in resource-limited hospital settings using an extended TAM–TOE framework. The topic is highly relevant to health informatics and health system strengthening, particularly in low- and middle-income countries undergoing digital transformation. The statistical analysis is generally rigorous; the use of PLS-SEM is appropriate for the complexity of the model and the sample size, and the authors provide a comprehensive evaluation of measurement reliability and validity.

At the same time, several methodological and contextual clarifications would enhance the transparency and interpretability of the study.

1. Contextual detail regarding hospital types and EHIS systems

The manuscript notes that participants were recruited from both governmental and nongovernmental hospitals, but additional information regarding the number, size, digital maturity, and geographical distribution of these facilities would help readers better understand the setting. It would also be helpful to clarify whether the participating hospitals used a unified EHIS platform or whether system characteristics varied across sites. Such information would strengthen the contextual grounding of the findings.

2. Departmental or unit-level distribution of participants

Although the study distinguishes between clinical and nonclinical roles, it does not report respondent distribution across specific departments (e.g., emergency, outpatient clinics, inpatient wards, laboratory, radiology, pharmacy). Given that workflow integration and system usage patterns differ substantially across units, providing this information—or acknowledging its absence—would help contextualize variations in acceptance.

3. Connection between empirical findings and practical implications

While the study identifies several strong predictors of EHIS acceptance (e.g., system quality, competitive pressure, PEOU, PU), some recommendations in the Implications section appear somewhat general and not fully aligned with the empirical results. For instance, competitive pressure showed a relatively strong effect, yet the discussion does not elaborate on how such dynamics manifest in the Palestinian hospital context. Strengthening this linkage would improve the relevance and specificity of the practical implications.

Recommendation: Minor Revision

Overall, this is a well-designed study with strong methodological foundations and meaningful contributions to the literature. The issues noted above primarily require clarification rather than substantial reanalysis. Therefore, I recommend Minor Revision, with attention to additional contextual detail and strengthened interpretation to maximize the clarity and utility of the manuscript.

Reviewer #4: Manuscript Number: PDIG-D-25-00865

Title: Adopting and validating a technology acceptance model-based paradigm to assess acceptance and satisfaction with electronic health information system by healthcare providers in resource-limited governmental and nongovernmental hospitals

Recommendation: Major Revision

Reviewer Summary

This manuscript presents a study validating an extended Technology Acceptance Model (TAM) integrated with the Technology-Organization-Environment (TOE) framework to assess the adoption of Electronic Health Information Systems (EHIS) in Palestinian hospitals. The study addresses a significant gap in the literature regarding digital health implementation in resource-constrained environments. The authors are to be commended for conducting rigorous data collection in a challenging setting. The choice of Partial Least Squares Structural Equation Modeling (PLS-SEM) is methodologically appropriate for the exploratory nature of the model, and the statistical reporting of validity metrics is generally robust.

However, there are substantial concerns regarding the interpretation of null results, the transparency of the sampling procedure, and the structural organization of the manuscript. Specifically, the conclusion that a non-significant finding implies "user resilience" is methodologically unsound and requires revision. I recommend Major Revisions to address the following points.

Major Comments

1. Clarification of Sampling Procedure and Response Rate

The manuscript states that the required sample size was calculated to be exactly 220 participants using the Raosoft calculator4. The Results section subsequently reports that exactly 220 healthcare providers participated in the study5. It is statistically unusual for the final number of valid, analyzed responses to match the calculated minimum threshold exactly, unless a specific stopping rule was applied or the response rate was 100%. To ensure reproducibility and transparency regarding potential selection bias, please clarify the data collection process. Specifically, did data collection cease immediately once the target of N=220 was reached, and were there any incomplete or excluded surveys? Please also report the total number of healthcare providers approached versus those who participated (i.e., the response rate).

2. Interpretation of Null Results regarding Complexity (CPLX)

In the Conclusion, the manuscript states that the non-significant effect of Complexity on Usefulness (P=.46) "highlight[s] the resilience of user attitudes". Drawing a specific psychological conclusion, such as user resilience, from a null statistical result is methodologically unsound. A failure to reject the null hypothesis indicates an absence of evidence for the relationship in this specific sample; it does not empirically validate the presence of "resilience," a construct that was not measured in the model. Please remove this speculative claim. Instead, discuss this null result in the context of the sample demographics. As shown in Table 1, participants reported a mean of 10.4 years of computer/EHIS usage. It is highly probable that this extensive experience—rather than "resilience"—mitigated the perceived effect of system complexity.

3. Substantiation of the "Gap" in Related Work

In the "Related Work" section, the manuscript states that hybrid TAM-TOE frameworks "tend to underperform" when deployed in resource-limited environments8. This assertion is the central justification for developing the extended model used in this study. However, the text does not currently provide specific comparative evidence to support this claim, such as citing lower R^2 values from specific previous studies in low-income versus high-income settings. Please strengthen the rationale for the study by citing specific examples or metrics that demonstrate this "underperformance." Providing concrete evidence that previous models fail to capture sufficient variance in developing contexts would make the argument for your extended model much more compelling.

4. Structural Coherence of the Introduction and Background The current organization of the opening sections disrupts the narrative flow of the manuscript. The text presents a "Background" section followed by a separate "Introduction" header, which creates redundancy. Furthermore, the narrative in these sections occasionally relies on emotive or "journalistic" phrasing (e.g., "disappointingly low", "falls short") rather than precise, objective descriptions. Please merge the "Background" and "Introduction" sections into a single, cohesive introductory section. Additionally, please refine the language to be more objective, providing specific adoption percentages or citation-backed data rather than qualitative descriptors.

Minor Comments

5. Causal Language in a Cross-Sectional Design The authors correctly acknowledge in the Limitations section that the cross-sectional design precludes causal inferences. However, the language used throughout the Abstract and Results sections is frequently deterministic, using terms such as "SQ enhanced PU," "CP drove PU," and "impact of...". Given that PLS-SEM on cross-sectional data establishes predictive association rather than causality, please revise these verbs to reflect association (e.g., "was associated with," "predicted," "was related to") to ensure scientific accuracy.

6. Model Parsimony and Theoretical Overlap The proposed research model is extensive, incorporating seven distinct exogenous predictors and 15 hypothesized paths. While the statistical discriminant validity appears adequate, there is significant theoretical overlap between constructs such as Relative Advantage (RA) versus System Quality (SQ), or Compatibility (COMP) versus Perceived Ease of Use (PEOU). Please expand the Discussion section to address the limitation of using such a high-dimensional model. Specifically, discuss whether the inclusion of so many predictors might dilute the explanatory power of individual factors or obscure the most critical drivers of adoption. Acknowledging the trade-off between model complexity and parsimony would strengthen the manuscript.

**Do you want your identity to be public for this peer review?** For information about this choice, including consent withdrawal, please see our Privacy Policy..

Reviewer #1: **Yes:** Nadia JelaniNadia JelaniNadia JelaniNadia Jelani

Reviewer #2: No

Reviewer #3: No

Reviewer #4: **Yes:** Dian HuDian HuDian HuDian Hu

**Figure resubmission:** While revising your submission, we strongly recommend that you use PLOS’s NAAS tool (https://ngplosjournals.pagemajik.ai/artanalysis) to test your figure files. NAAS can convert your figure files to the TIFF file type and meet basic requirements (such as print size, resolution), or provide you with a report on issues that do not meet our requirements and that NAAS cannot fix.

**Reproducibility:** To enhance the reproducibility of your results, we recommend that authors of applicable studies deposit laboratory protocols in protocols.io, where a protocol can be assigned its own identifier (DOI) such that it can be cited independently in the future. Additionally, PLOS ONE offers an option to publish peer-reviewed clinical study protocols. Read more information on sharing protocols at https://plos.org/protocols?utm_medium=editorial-email&utm_source=authorletters&utm_campaign=protocols To enhance the reproducibility of your results, we recommend that authors of applicable studies deposit laboratory protocols in protocols.io, where a protocol can be assigned its own identifier (DOI) such that it can be cited independently in the future. Additionally, PLOS ONE offers an option to publish peer-reviewed clinical study protocols. Read more information on sharing protocols at https://plos.org/protocols?utm_medium=editorial-email&utm_source=authorletters&utm_campaign=protocols

---

## [Decision Letter · Decision Letter 1]

18 Mar 2026

Adopting and validating a technology acceptance model-based paradigm to assess acceptance and satisfaction with electronic health information system by healthcare providers in resource-limited governmental and nongovernmental hospitals

PDIG-D-25-00865R1

Dear Prof shawahna,

We are pleased to inform you that your manuscript 'Adopting and validating a technology acceptance model-based paradigm to assess acceptance and satisfaction with electronic health information system by healthcare providers in resource-limited governmental and nongovernmental hospitals' has been provisionally accepted for publication in PLOS Digital Health.

Best regards,

Phat Kim Huynh, Ph.D.

Guest Editor

PLOS Digital Health

**Additional Editor Comments (if provided):**

**Reviewer Comments (if any, and for reference):**

Reviewer's Responses to Questions

**Comments to the Author**

Reviewer #1: All comments have been addressed

Reviewer #3: (No Response)

Reviewer #4: All comments have been addressed

publication criteria? Is the manuscript technically sound, and do the data support the conclusions? The manuscript must describe methodologically and ethically rigorous research with conclusions that are appropriately drawn based on the data presented.? Is the manuscript technically sound, and do the data support the conclusions? The manuscript must describe methodologically and ethically rigorous research with conclusions that are appropriately drawn based on the data presented.

Reviewer #1: Yes

Reviewer #3: (No Response)

Reviewer #4: Yes

3. Has the statistical analysis been performed appropriately and rigorously?

Reviewer #1: Yes

Reviewer #3: Yes

Reviewer #4: Yes

4. Have the authors made all data underlying the findings in their manuscript fully available (please refer to the Data Availability Statement at the start of the manuscript PDF file)?

The PLOS Data policy requires authors to make all data underlying the findings described in their manuscript fully available without restriction, with rare exception. The data should be provided as part of the manuscript or its supporting information, or deposited to a public repository. For example, in addition to summary statistics, the data points behind means, medians and variance measures should be available. If there are restrictions on publicly sharing data—e.g. participant privacy or use of data from a third party—those must be specified.requires authors to make all data underlying the findings described in their manuscript fully available without restriction, with rare exception. The data should be provided as part of the manuscript or its supporting information, or deposited to a public repository. For example, in addition to summary statistics, the data points behind means, medians and variance measures should be available. If there are restrictions on publicly sharing data—e.g. participant privacy or use of data from a third party—those must be specified.

Reviewer #1: Yes

Reviewer #3: Yes

Reviewer #4: Yes

5. Is the manuscript presented in an intelligible fashion and written in standard English?

Reviewer #1: Yes

Reviewer #3: Yes

Reviewer #4: Yes

Reviewer #1: The authors have thoroughly addressed the concerns raised in the previous round of review. In particular, the dedicated discussion of the unexpected structural paths (e.g., negative Compatibility → Perceived Usefulness and the non-significant TMS/ITS → PEOU paths) substantially strengthens the manuscript’s theoretical contribution. The clarification of sampling procedures, inclusion of structural diagrams and hypothesis summary tables, and compliance with PLOS data availability requirements improve methodological transparency.

The manuscript is now technically sound, theoretically coherent, and suitable for publication. I recommend acceptance.

Reviewer #3: (No Response)

Reviewer #4: This revision has addressed all points raised by the reviewers faithfully, therefore I support the publication of this paper.

**Do you want your identity to be public for this peer review?** For information about this choice, including consent withdrawal, please see our Privacy Policy..

Reviewer #1: No

Reviewer #3: No

Reviewer #4: **Yes:** Dian HuDian HuDian HuDian Hu
